# Characterization of *Fusarium* Diversity and Head Microbiota Associated with Rice Spikelet Rot Disease

**DOI:** 10.3390/plants14101531

**Published:** 2025-05-20

**Authors:** Qun Cao, Jianyan Wu, Tianling Ma, Chengxin Mao, Chuanqing Zhang

**Affiliations:** College of Advanced Agricultural Sciences, Zhejiang Agriculture and Forest University, Hangzhou 311300, China; 13052945107cq@sina.com (Q.C.); wujianyan@zafu.edu.cn (J.W.); czipotw@163.com (T.M.); zafumaocx@163.com (C.M.)

**Keywords:** RSRD, *Fusarium* spp., *Curvularia* spp., amplicon sequencing, multi-locus phylogenetic analysis

## Abstract

Rice spikelet rot disease (RSRD) affects the production and market price of rice, and can be harmful to humans and livestock. In this study, 51 strains of *Fusarium* spp. were isolated from rice spikelets in the Lin’an, Yuhang, and Fuyang regions. The isolates comprised four composite species: *Fusarium incarnatum-equiseti* species complex (FIESC), *Fusarium asiaticum* species complex (FSAMSC), *Fusarium fujikuroi* species complex (FFSC), *Fusarium commune* species complex (FNSC), and five species of *Fusarium* spp. (*F. luffae*, *F. sulawesiense*, *F. asiaticum*, *F. fujikuroi*, and *F. commune*). The separation rate of *F. sulawesiense* was the highest (41.2%), followed by *F. asiaticum* at 37.3%. The results of this study, compared with those of other studies, found that the newly discovered species of *Fusarium* spp. associated with RSRD were FSAMSC (*F. asiaticum*) and FNSC (*F. commune*). Temperature is one of the important factors causing RSRD; the optimal growth temperature for *F. sulawesiense* and *F. commune* was 30 °C, and the optimal growth temperature for other species was 25 °C. A high temperature of 35 °C did not inhibit the growth of *Fusarium*, as *F. commune* and *F. fujikuroi* could grow at this temperature. At 20–30 °C, the growth rate of *F. asiaticum* was higher than those of other strains. To determine whether the occurrence of RSRD is related to the correlation between *Fusarium* and other fungi in rice spikelets under different health conditions, the genetic diversity of fungi in rice spikelets was analyzed by amplicon Internal Transcribed Spacer (ITS) sequencing, and the correlations between strains of *Fusarium* spp. were measured. The results showed that the fungal diversity of diseased rice spikelets (RD) was higher than that of relatively healthy rice spikelets (RH). *Curvularia* spp., which was affected by the condition of the spikelets, was negatively correlated with *Fusarium* spp. in RH and positively correlated with *Fusarium* spp. in RD. Therefore, *Fusarium* spp. and *Curvularia* spp. can jointly cause the occurrence of RSRD. The results of the study are significant for understanding the occurrence of RSRD and formulating prevention and control measures.

## 1. Introduction

Rice (*Oryza sativa* L.), as the primary source of carbohydrates for billions of people worldwide, holds significant importance in both economic and political contexts [1]. The frequent occurrence of rice diseases affects both rice yield and quality, thus seriously threatening the food supply on a global scale. Rice spikelet rot disease (RSRD) has become another major threat to rice in China [1]. RSRD is found in the main rice-growing areas of China, and the initial symptoms of the disease are characterized by reddish-brown or rust-red spots on the flowering flowers, which turn black-brown at the ripening stage of rice [2]. The affected spikelets are partially full or aborted, unable to produce edible rice grains, or causing discoloration, incompleteness, and deformation of rice grains. RSRD can be caused by *Fusarium* spp., *Curvularia* spp., *Bipolar* spp., *Aspergillus* spp., and *Alternaria* spp. [1,3]. Domestic and foreign research has shown that *Fusarium* species are the main pathogenic fungi.

*Fusarium* spp. are important pathogens of cereal crops. The genus comprises several economically important fungal plant pathogens [4] and producers of mycotoxins [5]. *Fusarium* spp. is known to cause root and stem rot [6], canker disease, vascular wilt, fruit and seed rot, and leaf diseases. *Fusarium* spp. affects major food and cash crops such as wheat, rice, maize, bananas, and cotton, often with devastating effects on production and management [7]. As one of the most destructive fungal pathogens of rice, *Fusarium* spp. can infect the plants at the ear and flowering stages, causing grain deterioration, deformity, and mildew, collectively referred to as RSRD [7]. According to domestic research, the species of *Fusarium* spp. that cause RSRD are as follows: In the northeast rice region, the percentages of *Fusarium incarnatum-equiseti* species complex (FIESC), *F. tanahbumbuense*, *F. proliferatum*, *F. verticillioides*, and *F. graminearum* are 43, 35, 13, 4, and 4%, respectively. In the southern rice region, the percentages for *F. graminearum*, FIESC (Fujian), FIESC, *F. verticillioides*, and *F. tanahbumbuense* (Hainan) are 58, 25, 26, 9, and 9%, respectively. In the central rice region, the corresponding percentages for FIESC, *F. tanahbumbuense*, *F. fujikuroi* (Anhui), and FIESC (Jiangsu) are 69, 9, 3, and 40%, respectively [3]. There is a report of RSRD being caused by *F. sulawesiense* in India [8], according to which RSRD not only affects the yield of rice but also alters the appearance of rice grains and reduces the quality of the rice due to the colored pathogenic fungi. Importantly, RSRD often occurs in the heading and flowering stages of rice, causing rice grain spoilage, deformity, and mildew, as well as the accumulation of mycotoxins in the grains [7]. For example, nivalenol (NIV) and its acetylated derivative (4-acetylnivalenol; 4-ANIV), diacetoxyscirpenol (DAS), neosolaniol (NEO), zearalenone (ZEN), and beauvericin (BEA) are produced by the FIESC; NIV, 4-ANIV, and 3-acetyldeoxynivalenol (3-ADON) are produced by *F. asiaticum* [5]. Therefore, RSRD not only affects rice production and market price but also poses a threat to human and animal health and safety [9].

The importance of interactions between plant pathogens and the plant microbiome has been increasingly recognized. Plant-associated microorganisms may also facilitate infection by plant pathogens, while beneficial microbial strains can ameliorate plant diseases [10,11]. *Microdochium* spp. often co-occur with *Fusarium* spp. and can cause head blight [10]. Root symbiotic B3 can significantly reduce the incidence of RSRD and the production of fumonisin by *F. proliferatum* [2]. In Fusarium head blight of wheat (FHB), interactions between species that cause the disease and other members of the plant microbiome play an important role in disease outbreaks and mycotoxin accumulation in the grains [10]. Therefore, the microbiome plays an important role in plant disease control.

At present, the mechanism of RSRD and the interaction between fungal communities and Fusarium at the time of infection are not clear. Therefore, this study analyzed the distribution of fungi in diseased and relatively healthy rice ears from the perspective of the microbiome, as well as the correlations between *Fusarium*, the main pathogen causing RSRD, and other fungi. In order to clarify the *Fusarium* spp. causing RSRD in Zhejiang Province, we employed the multi-site tandem method to identify the *Fusarium* species causing RSRD in the Hangzhou, Zhejiang Province region of China [12]. The aims of this study were to examine the mechanism of RSRD, the distribution of fungal diversity, the correlations between *Fusarium* and other fungi causing RSRD, and the influence of *Fusarium* diversity and temperature on RSRD.

## 2. Results

### 2.1. Composition of Diseased and Relatively Healthy Rice Spikelet Fungal Species

Amplicon sequencing was performed on diseased and relatively healthy rice spikelets to reflect the species changes among samples. Species classification of OTUs was carried out by comparisons with public databases, and the relative abundances of fungal species in rice ears in different health conditions were determined at subordinate levels. The results of this study showed that the following 22 genera were found in both relatively healthy rice ears (RH) and diseased rice ears (RD), and the relative abundances were (RH vs. RD), respectively, *Curvularia* spp. (0.011 vs. 0.012), *Ustilaginoidea* spp. (0.041 vs. 0.049), *Pyrenochaetopsis* spp. (0.002 vs. 0.023), *Cladosporium* spp. (0.008 vs. 0.005), *Nigrospora* spp. (0.34 vs. 0.24), *Coprinopsis* spp. (0.034 vs. 0.001), *Violaceomyces* spp. (0.045 vs. 0.001), *Pyxidiophorales*_gen (0.001 vs. 0.019), *Myrmecridium* spp. (0.001 vs. 0.042), *Candida* spp. (0.010 vs. 0.008), *Phaeosphaeria* spp. (0.008 vs. 0.014), *Mucor* spp. (0.001 vs. 0.013), *Penicillium* spp. (0.060 vs. 0.008), *Ophiosphaerella* spp. (0.005 vs. 0.014), *Fusarium* spp. (0.015 vs. 0.129), *Clonostachys* spp. (0.011 vs. 0.021), *Agaricus* spp. (0.014 vs. 0.002), *Sarocladium* spp. (0.052 vs. 0.144), *Aspergillus* spp. (0.038 vs. 0.012), *Chaetomella* spp. (0.001 vs. 0.021), *Naganishia* spp. (0.010 vs. 0.010), and *Neosetophoma* spp. (0.038 vs. 0.023). RD led to an increase in the relative abundance of *Ustilaginoidea* spp., *Pyrenochaetopsis* spp., *Pyxidiophorales*_gen, *Myrmecridium* spp., *Phaeosphaeria* spp., *Mucor* spp., *Ophiosphaerella* spp., *Fusarium* spp., *Clonostachys* spp., *Sarocladium* spp., and *Chaetomella* spp. (Figure 1).

### 2.2. Fusarium Correlated with Other Fungal Groups

Pearson correlation was used to analyze the correlations between *Fusarium* spp. and other fungi in rice spikelets of different states (Figure 2). The results showed that eight strains of fungi in RH were negative correlated with *Fusarium* spp., and their Pearson correlation coefficients were as follows: *Phaeosphaeria* spp. −0.4, *Pyrenochaetopsis* spp. −0.37, *Sarocladium* spp. −0.34, *Ophiosphaerella* spp. −0.34, *Curvularia* spp. −0.33, *Cladosporium* spp. −0.32, *Neosetophoma* spp. −0.32, and *Clonostachys* spp. −0.31 (Figure 2a,b). Four strains of fungi in RD were positively correlated with Fusarium, with Pearson coefficients of *Curvularia* spp. 0.87, *Pyxidiophorales* spp. 0.69, *Penicillium* spp. 0.54, *Sarocladium* spp. 0.47, and *Clonostachys* spp. 0.43. In RD, *Fusarium* spp. is highly correlated with *Curvularia* spp. The relative abundance of *Fusarium* spp. in RD increased significantly, which was related to *Curvularia* spp. (Figure 2c,d).

### 2.3. Fusarium Isolates

A total of 60 diseased rice spikelets (RD) were collected from Yuhang, Lin’an, and Fuyang counties and used for fungal isolation. Of these, 51 Fusarium isolates were identified based on colony morphology and tef1 gene sequences. The tef1 gene sequence from each Fusarium isolate was amplified with a 100% success rate, and the sequences were blasted against the Fusarioid-ID database. Based on >99% similarity, the rice spikelets were contaminated with four species complexes (Appendix A): FIESC (29 isolates, 56.9%), FSAMSC (19 isolates, 37.2%), FFSC (two isolates, 3.9%), and FNSC (one isolate, 2%). Among the four species complexes, FIESC isolates were the highest in frequency in each county, followed by FSAMSC isolates (Appendix A). We used multi-locus phylogenetic analyses to identify the FIESC and FSAMSC species to further resolve Fusarium populations infecting rice spikelets in the field (Figure 3).

### 2.4. Multi-Locus Phylogenetic Analysis

For the FIESC isolates, 29 isolates together with 33 reference isolates (Appendix A), including the outgroup *F. concolor* (NRRL 13459), were employed in a multi-locus phylogenetic analyses of three gene sequences (Table 1). The concatenated sequences comprised an alignment of translation elongation factor 1-alpha (*tef1*), RNA polymerase largest subunit (*rpb1*), and RNA polymerase second largest subunit (*rpb2*). The results showed that 29 isolates clustered with two species: 22 isolates with *F. sulawesiense* (bootstrap = 92%) and 7 isolates with *F. luffae* (bootstrap = 99%) (Figure 4).

Nineteen FSAMSC isolates along with reference isolates, including the outgroup *F. kyushuense* (LC18277), were employed in a multi-locus phylogenetic analysis using concatenated sequences of histone (*H3*), *rpb1*, *rpb2*, and *tef1*. The phylogenetic tree showed that all 19 field isolates were clustered with *F. asiaticum* (bootstrap = 94%) (Figure 5).

The FFSC and FNSC isolates used to construct the phylogenetic tree were subjected to multi-locus phylogenetic analyses with concatenated sequences of calmodulin (*CaM*), *rpb1*, *rpb2*, *tef1*, and beta-tubulin (*tub2*) for FFSC and *tef1*, *rpb1*, and *rpb2* for FNSC. The results showed that two isolates (YR05 and YR15) were clustered with *F. fujikuroi* (bootstrap = 84%) (Figure 6), and isolate YR02 was clustered with *F. commune* (bootstrap = 100%) (Figure 7).

### 2.5. Morphological Characteristics

Based on phylogenetic tree identification results, morphological observation of five species of *Fusarium* isolated from rice spikelets in three regions was conducted (Table 2). Preliminary morphological observation showed that *F. sulawesiense* and *F. luffae* were morphologically similar (Figure 8 and Figure 9).

*F. sulawesiense* (Figure 8, Table 2): Colonies of isolate LA03 grown on PDA in the dark reached 25–26 mm diameter after 3 days at 25 °C, with villous colonies, dense aerial mycelia, and an undulate colony margin, and were white; they were reverse yellowish grey in the center, and white at the margin. Colonies grown on OA in the dark reached 40–42 mm diameter after 3 days at 25 °C, and were flat, radially striated, with entire margins, and white and reverse white. Colonies grown on SNA in the dark reached 27–30 mm in diameter after 3 days at 25 °C, and were flat, with scant aerial mycelia, and without regular colony margins; they were reverse white (Figure 8a1–a4). Sporodochia macroconidia were falcate and slender, with both ends slightly curved dorsiventrally, a conical to slightly papillate apical cell, and a well-developed, foot-shaped basal cell, and the hyaline was thin and smooth-walled, and 3–5-septate (Figure 8a5,a6).

*F. luffae* (Figure 9, Table 2): Colonies of isolate LA01 grown on PDA in the dark reached 35–39.5 mm diameter after 3 days at 25 °C. Colonies were villous, with fluffy aerial mycelia, and an undulate colony margin, and were white; they were reverse pale orange, and white at the margin. Colonies grown on OA in the dark reached 50–52 mm diameter after 3 days at 25 °C. Colonies were flat, and radially striated; the colony had entire margins, and were white and reverse white (Figure 9b1–b4). Colonies grown on SNA in the dark reached 30–33 mm diameter after 3 days at 25 °C. Colonies were flat, and the aerial mycelia scant, without regular colony margins; they were reverse white. They had an apricot pigment on PDA (Figure 9b2,b5). Sporodochial conidia were falcate, slightly slender, straight to slightly curved dorsiventrally, with a conical to slightly papillate apical cell and a well-developed and foot-shaped basal cell, and the hyaline was thin and smooth-walled, and 3–5-septate (Figure 9b5,b6).

*F. asiaticum* (Figure 10, Table 2): Colonies of isolate LA02 grown on PDA in the dark reached 76–78.5 mm diameter after 3 days at 25 °C. Colonies were villous, with fluffy aerial mycelia, and entire colony margins, and were white on PDA. Colonies grown on OA in the dark reached 55–58 mm diameter after 3 days at 25 °C. Colonies were flat and radially striated, with entire colony margins. Colonies grown on SNA in the dark reached 36–40 mm diameter after 3 days at 25 °C. Colonies were flat and white, with scant aerial mycelia and erose margins (Figure 10c1–c3). They had a pink pigment on PDA, OA, and SNA (Figure 10c1–c3). Sporangium conidia were falcate, dorsalis curved, with conical apex cells and well-developed foot basal cells, and were transparent, thin-walled smooth, and 1–6-septate.

*F. fujikuroi* (Figure 11, Table 2): Colonies of isolate YR15 grown on PDA in the dark reached 61–62 mm diameter after 5 days at 25 °C. Colonies showed a spider network; air mycelium developed, and the colony edges were neat and white, and the backs were yellow. Colonies grown on OA in the dark reached 61–62 mm diameter after 5 days at 25 °C. Colonies were flat, with radial stripes and whole colony margins, and were white. Colonies grown on SNA in the dark reached 52–54 mm diameter after 5 days at 25 °C. Colonies were flat; the air mycelia were dense, and the colony edges were flat and white (Figure 11d1–d3). They had an apricot pigment on PDA, which was absent on OA and SNA (Figure 11d1–d3). Sporangium conidia were oval, with rounded apex cells and well-developed foot base cells, and were transparent, thin-walled, and smooth (Figure 11d4).

*F. commune* (Figure 12, Table 2): Colonies of isolate YR02 grown on PDA in the dark reached 73–75 mm diameter after 5 days at 25 °C. Colonies were woolly, with fluffy air mycelia, and colony margins which were whole and white. The backs of colonies were yellow. Colonies grown on OA in the dark reached 62–63 mm diameter after 5 days at 25 °C. Colonies were flat, with radial stripes; colony margins were whole and white. Colonies grown on SNA in the dark reached 60–61 mm diameter after 5 days at 25 °C. Colonies were flat; the air mycelia were dense, and the colony edges were flat and white (Figure 12e1–e3). They had an apricot pigment on PDA, which was absent on OA and SNA (Figure 12e1–e3). Sporangium conidium were falcate, dorsalis curved, with conical and bluntly rounded apex cells and well-developed foot basal cells, and were transparent, thin-walled smooth, and without septa or with 1–3 septa (Figure e4).

### 2.6. Effect of Temperature on Growth, Spore Production, and Spore Germination

When rice is in the booting stage to flowering stage and overlapped with cloudy, rainy (high humidity), and warm (25–33 °C) conditions, rice panicle rot disease will occur. Five isolated *Fusarium* species, *F. luffae*, *F. asiaticum*, *F. sulawesiense*, *F. commune*, and *F. fujikuroi*, were tested for the effect of temperature on growth (Figure 13). The optimal growth temperature for *F. sulawesiense* and *F. commune* was 30 °C, and the optimal growth temperature for other species was 25 °C. The highest temperature inhibited the growth of *Fusarium*, as only *F. commune* and *F. fujikuroi* were able to grow at 35 °C. At 20–30 °C, the growth rate of *F. asiaticum* was higher than those of other strains. At 20 °C, there was no significant difference in growth rate between *F. sulawesiense* and *F. luffae* (Figure 13a). Spore suspensions were prepared by incubating *Fusarium* from rice spikelets at 20, 25, 30, and 35 °C for 7 days, and the effects of different temperatures on the sporulation ability of *Fusarium* were observed. The results showed that *F. sulawesiense*, *F. commune*, and *F. fujikuroi* could produce spores at different temperatures. However, there were significant differences in the quantity of spores produced at different temperatures. *F. luffae* and *F. asiaticum* did not produce spores at 35 °C, while the optimal sporulation temperature for *F. sulawesiense* was 35 °C. The optimal sporulation temperature for *F. luffae*, *F. asiaticum*, *F. commune*, and *F. fujikuroi* was 30 °C (Figure 13c). Experiments on the spore germination of *Fusarium* of RSRD were performed at different temperatures. The results showed that *F. asiaticum* produced more spores than other species. The spores of *F. sulawesiense*, *F. luffae*, and *F. fujikuroi* did not germinate at 35 °C. A temperature of at 30 °C was optimal for spore germination of *F. luffae*, *F. sulawesiense*, and *F. fujikuroi*. There was no significant difference in the number of spores produced between *F. commune* at the four different temperatures (Figure 13b).

### 2.7. Pathogenicity Test

After spraying with the spore suspension for 7 days, the results showed that each pathogen could cause disease on rice spikelets. Spikelets sprayed with conidial suspensions of any of the five pathogens developed typical RSRD symptoms. The rice grains were discolored, with white and gray mildew layers (Figure 14).

## 3. Discussion

In recent years, rice spikelet rot disease (RSRD), as an emerging rice disease, has affected rice production and management in rice-growing areas of China [13]. Pathogenic *Fusarium* spp. are currently regarded as the main cause of sheath rot disease, However, reports of *Fusarium* associated with RSRD remain limited [14]. In order to better identify *Fusarium* species, Han et al. identified different DNA-coding genes in tandem according to different composite species, for example, calmodulin (*CaM*), translation elongation factor 1-alpha (*tef1*), RNA polymerase largest subunit (*rpb1*), RNA polymerase second largest subunit (*rpb2*), and beta-tubulin (*tub2*) for *Fusarium fujikuroi* species complex (FFSC); for *Fusarium incarnatum-equiseti* species complex (FIESC) and *Fusarium nisikadoi* species complex (FNSC), *rpb1*, *rpb2*, and *tef1*; and for *Fusarium asiaticum* species complex (FSAMSC), histone (*H3*), *rpb1*, *rpb2*, and *tef1* [12]. Using the above methods, in this study, five species of *Fusarium*, including *F. sulawesiense*, *F. asiaticum*, *F. luffae*, *F. commune*, and *F. fujikuroi*, were isolated and identified, among which *F. asiaticum*, *F. luffae*, and *F. commune* were firstly determined as pathogens of RSRD. Previous research has indicated that disease occurrence and epidemics occur when rice has a late booting to flowering period overlapping with overcast, rainy (high humidity), and warm (25–33 °C) conditions [15]. The results of this study showed that *Fusarium* causing RSRD had broad adaptability to temperature, especially *F. commune* and *F. fujikuroi*.

Previous research has shown that piercing–sucking insects may facilitate the direct introduction of *Fusarium* into plant tissues [2,16], while the relationship between RSRD and *Fusarium* is not well understood. There are few reports on the relationship between RSRD and the diversity of rice panicle fungi. Therefore, we compared the fungal diversity of RH and RD grains sampled from the field. Previous studies have reported that diseased samples had greater microbial diversity than healthy samples [16]. We found that the fungal diversity in RD samples was significantly higher than that in RH samples. A correlation analysis showed that *Fusarium* and *Curvularia* were positively correlated. In addition to the relationship with other fungi, *Fusarium* and *Curvularia* were associated with the formation of RSRD. According to previous research, the microorganisms associated with rice ear diseases are very diverse [17]. Interestingly, we found a significant increase in the relative abundance of the beneficial fungi *Myrecrisium* and *Chaetomella* in RD, perhaps due to the fact that when plants are under stress, beneficial microorganisms can be recruited from the environment through “cry for help” strategies to respond to the stress in a coordinated manner [18]. In the correlation analysis with *Fusarium*, *Fusarium* and *Curivularia* were highly correlated with *Curivularia* in RD. In the study of Huang et al., *Fusarium* and *Curivularia* could co-infect rice to cause RSRD [1], so it is necessary to pay attention to infection with *Curivularia* while paying attention to *Fusarium* in RSRD. Although this study identified the differences in the relative abundances of different fungi in rice panicles under different healthy states by analyzing the fungal communities, and the high correlation between *Fusarium* and *Curivularia*, further research is needed on how the whole fungal community interacts (Figure 15).

As the main pathogen of RSRD, the diversity of *Fusarium* in the Hangzhou area is different from that in other regions. We found that high temperatures do not completely inhibit the growth of the pathogen. Therefore, when controlling RSRD, targeted measures should be based on the types and characteristics of local pathogens. According to previous studies, the microorganisms associated with rice spike diseases are very diverse. The microbial diversity in diseased rice spikes (RD) is higher than that in healthy ones. We found that *Fusarium* spp. and *Curvularia* spp. are positively correlated in RD; however, it is unclear how *Fusarium* spp. and *Curvularia* spp. interact and what role they play in the overall fungal community that causes RSRD. In the future, we can explore the interactions between fungal communities in diseased rice spikes from a microbial perspective.

## 4. Materials and Methods

### 4.1. Media

The media used in this research are shown in Table 3.

### 4.2. Sample Collection, Processing, and Preservation

In July 2023, rice fields were investigated in the Yuhang, Lin’an, and Fuyang Districts of Hangzhou City, Zhejiang Province, China. Rice spikelet samples were taken from the paddy fields in the above three areas. The sampling strategy was as follows: two rice ears were taken from each of the above three areas, each field was sampled by the five-point sampling method, and four rice ears were selected from each point, which were RD and RH, respectively, and a total of 120 rice ears were taken (RH: 60 plants, RD: 60 plants). The samples were divided into two treatments (diseased rice spikelets: RD; and relatively healthy rice spikelets: RH) [12]. The incidence of diseased rice spikelets concentrated on rice spikelet rot (Figure 16). The rice spikelet samples were sent to Beijing Genomics Institute (BGI) for amplicon sequencing and pre-delivery treatment. Each sample was weighed (≥3 g); dust on the surface of the plants was washed with running water, and the plants were disinfected with sodium hypochlorite for 2 min. The samples were placed in sterile water, and two sterile steel balls were added to the samples and were shaken vigorously to remove bacteria on the surface of the rice spikelets. After the above treatment, the samples were placed in a cool and ventilated place to dry and stored at −20 °C.

### 4.3. Extraction of Microbiome DNA

DNA was extracted from samples using a MagPure Stool DNA KF Kit B (MAGEN, Guangzhou, China) and a Magnetic Bead Fecal and Soil Genome Extraction Kit (MAGEN, Guangzhou, China), according to the manufacturer’s instructions. Samples (100–200 mg) were transferred to a centrifuge tube containing grinding beads. After the addition of 1 mL of ATL/PVP-10 buffer, the samples were ground in a grinding machine (Shanghai Jingxin Tech, Shanghai, China) and incubated at 65 °C for 20 min. The mixture was centrifuged at 14,000× *g* for 5 min (Eppendorf, Oldenburg, Germany). The supernatant was transferred to a new tube, and 0.6 mL of PCI buffer were added to the sample and then mixed thoroughly by vortexing for 15 s. The mixture was centrifuged at 18,213× *g* for 10 min. The supernatant was transferred to a deep well plate with magnetic bead binding solution (600 μL of buffer with magnetic beads, 20 μL of Proteinase K, 5 μL of RNase A, 700 μL of Wash 1, 700 μL of Wash 2, 700 μL of Wash 3, and 100 μL of elution buffer). The sample was transferred to the corresponding place in a deep well plate of the machine (Kingfisher, Thermo Fisher, Shanghai, China) and the corresponding program was initiated. The DNA was transferred to a 1.5 mL centrifuge tube for storage following termination of the program. Before library preparation, the samples were thawed on ice, mixed well, centrifuged, and the appropriate amount of samples was placed on a Qubit fluorometer or microplate reader (Thermo Fisher Scientific Inc, Shanghai, China) for testing.

### 4.4. Meta Amplicon Library Preparation (Illumina)

A library was prepared using 2 × Phanta Max Master Mix (VAZYME, Nanjing, China) polymerase. PCR amplification was conducted using Internal Transcribed Spacer (ITS) rDNA primers, with positive and reverse F and R in the ITS1/ITS2 variable region (ITS1 region primers ITS1: CTTGGTCATTTAGAGGAAGTAA, and ITS2: GCTGCGTTCTTCATCGATGC; and ITS2 region primers ITS3: GCATCGATGAAGAACGCAGC, and ITS4: TCCTCCGCTTATTGATATGC). PCR enrichment was performed in a 50 μL reaction volume containing 30 ng of template and fusion PCR primers. The PCR cycling conditions were as follows: 95 °C for 3 min; and 30 cycles of 95 °C for 15 s, 56 °C for 15 s, and 72 °C for 45 s; and a final extension at 72 °C for 5 min. The PCR products were purified using DNA magnetic beads (BGI, LB00V60). The validated libraries were used for sequencing on an Illumina MiSeq platform (BGI, Shenzhen, China) following the standard Illumina pipelines, generating 2 × 250 paired-end reads.

### 4.5. Meta Amplicon Analysis

The raw data were filtered to generate high-quality clean reads as follows [20]. First, reads whose average Phred quality values were lower than 20 over a 30 bp sliding window were truncated. Reads whose lengths were 75% of their original length after truncation were removed. Reads that were contaminated by adapter sequences and those with ambiguous bases (N bases) were removed, along with low-complexity reads. The software used for quality control was iTools Fqtools fqcheck v.0.25. Adapters were removed using cutadapt v.2.6. Filtering was performed using readfq v1.0. For tag connections and operational taxonomic unit (OTU) clustering, the Usearch method was used for clustering, and sequence splicing was performed with Fast Length Adjustment of Short reads, v1.2.11 (FLASH) [20]. FLASH was also used to assemble the paired-end reads obtained by double-end sequencing through overlap relationships. The stitching conditions were as follows: minimum overlapping length = 15 bp; mismatch ratio of overlapped regions = 0.1. Tags were clustered to OTUs (Operational Taxonomic Units) with USEARCH v7.0.1090 [21] using a 97% threshold by UPARSE, where the unique OTU representative sequences were obtained. Chimeras were filtered by UCHIME v4.2.40 [21]. ITS used the existing chimera database (UNITE v201706 28) to remove chimeras by comparisons. The sequences were divided into ITS full-length, ITS1, and ITS2 and selected according to the sequencing region. The usearch_global method was used to compare all tags to the OTU representative sequences to obtain the OTU abundance statistics for each sample. OTU representative sequences were aligned against the database for taxonomic annotation by the RDP classifier v2.2 software, where the sequence identity was set to 0.6. The annotation results were filtered by removing OTUs that were not annotated, as well as taxa that did not match with the project’s research background.

### 4.6. OTU Taxonomic Annotation and Species Composition and Relative Abundance Analysis

OTU representative sequences were aligned against the database for taxonomic annotation by the RDP classifier v2.2 software. The sequence identity was set as 0.6. The annotation results were filtered as follows. OTUs that were not annotated and those that did not match the project’s research background were removed. The OTU annotation method was used to analyze the species composition and identify the changes and relative abundance of species composition among samples [22]. Species classification of OTUs was carried out by comparing with the ITS database, and the species abundance of each sample crop was analyzed at the phylum, class, order, family, genus, and species levels. Starting from the class level, the species in the group with an average abundance lower than 0.5% and all species not annotated at this classification level were combined in the BIG microbial amplicon analysis platform, Rv3.4.1. In this study, fungal species composition and the relative abundances of diseased and relatively healthy rice ears are demonstrated from a secondary perspective.

### 4.7. Correlation Analysis

A Pearson correlation analysis was carried out on the abundance values of *Fusarium* with those of other fungi in diseased and relatively healthy rice ears. The analysis was performed using the OmicStudio tools (https://www.omicstudio.cn/tool/62, accessed on 28 August 2024) [23]. The Shapiro–Wilk test was used to check the normality of the variables. A *p* value greater than 0.05 indicated that the data were normally distributed. *p* values were calculated using two-tailed tests [23].

### 4.8. Fusarium Isolation and Preservation

The *Fusarium* from rice ears was separated by direct mycelium isolation [24]. The detailed operation was as follows. The sampled rice ear was cut from the grain with scissors, and the dust on the sample surface was washed with water. The sample was then placed on absorbent paper to dry in a cool, ventilated place. The dried rice grains were soaked in 75% ethanol solution for 1 min, then soaked in 3% sodium hypochlorite solution for 3 min, and finally rinsed 3 times with sterile water. The rice grain tissue after sterilization was placed on sterile paper for static drying and then inoculated onto PDA medium containing streptomycin (five pieces per dish) and incubated at a constant temperature of 25 °C [25,26]. After 3 days, when the colonies had grown out, the mycelium of *Fusarium* was selected from the separation plate and transferred to PDA medium. The newly grown *Fusarium* was placed in a PDA frozen storage tube and stored in the refrigerator at 4 °C for subsequent tests.

### 4.9. DNA Extraction and Amplification

The purified Fusarium culture was activated on PDA medium; after 3 days, Fusarium DNA was extracted using a fungal gene DNA rapid extraction kit (Shengong Biology, Shanghai, China). The specific procedure was as follows. Fusarium mycelium was scraped and collected in a 2 mL aseptic centrifuge tube; two aseptic grinding beads (diameter 2 mm) and 400 μL of digestion buffer were added and then oscillated and mixed on a vortex instrument for grinding (the working parameters were room temperature, 120 s, and 60 Hz). The mixture was placed in a water bath at 65 °C for 1 h until the cells were completely dissolved. During this time, the mixture was placed upside down every 10 min. After 1 h, 200 μL of PF buffer were added, and the sample was thoroughly mixed upside down. The mixture was placed in the refrigerator at −20 °C for 5 min, followed by centrifugation at 10,000 rpm at room temperature for 5 min. Then, 500 μL of the supernatant were transferred to a new aseptic centrifuge tube; an equal volume of isopropyl alcohol was added, and the sample was inverted and thoroughly mixed 8 times, then kept at room temperature for 3 min. This was followed by centrifugation at 10,000 rpm for 5 min. The supernatant was discarded, and 1 mL of 75% ethanol was added and this was rinsed inverted for 3 min, then centrifuged at room temperature and 10,000 rpm for 2 min. The supernatant was discarded, and the sample was rinsed and centrifuged twice. The sample was then inverted for 10 min at room temperature until the residual ethanol had volatilized, then dissolved in 50 μL of TE buffer (10 mM Tris-HCl, 1mM EDTA, pH 8.0), and stored in the refrigerator at −20 °C.

*Fusarium* isolates from 60 infected rice ears were preliminarily identified based on colony morphology and tef1 sequences [12,27]. The identification of Fusarium strains required three steps: (1) the tef1 gene sequence of each *Fusarium* strain was amplified, and the success rate of amplification was 100%; (2) different Fusarium complex types were determined by the results of *tef1* sequence alignment [28,29]; and (3) phylogenetic analysis of Fusarium species complexes performed on several multi-locus datasets. This was conducted using calmodulin (*CaM*), RNA polymerase largest subunit (*rpb1*), *rpb2*, *tef1*, and β-tubulin (*tub2*) for FFSC [1]. FIESC and FNSC used *rpb1*, *rpb2* and *tef1*; and histones (*H3*), *rpb1*, *rpb2*, and *tef1* were used in FSAMSC [12,28,29]. After the above steps, 51 strains were obtained. The amplification procedures and primer information are listed in Table 4.

### 4.10. Multi-Locus Phylogenetic Analysis of Fusarium Associated with Rice Spikelet Rot

Once the sequencing results were complete, the *tef1* sequencing results were used for comparisons in the NCBI database (https://www.ncbi.nlm.nih.gov accessed on 23 August 2024) and the *Fusarium* MLST database (https://fusarium.mycobank.org accessed on 30 August 2024). Subsequently, phylogenetic trees were constructed according to multiple loci tandem sequences of different composite species [12]. The results of composite species comparisons are shown in Appendix A. Based on the above *tef1* comparison results, phylogenetic analyses of different *Fusarium* species complexes were performed using different multi-locus datasets. *CaM*, *rpb1*, *rpb2*, *tef1*, and *tub2* were used for FFSC; *rpb1*, *rpb2*, and *tef1* were used for FIESC and FNSC; and *H3*, *rpb1*, *rpb2*, and *tef1* were used for FSAMSC [12,28,29]. Information concerning the strains used in this study for the construction of the gene evolution tree is listed in Appendix A.

The sequences of the strains were searched in the nucleic acid sequence database of the National Center of Biotechnology Information (NCBI) to determine the selection of reference strains. Each gene entry number of the reference strain was obtained through a literature search, and its sequence information was retrieved and downloaded from GeneBank. The sequences of the standards and isolates were aligned and corrected in Fasta format by the MEGA 7.0 software [25]. The corrected individual gene sequences were concatenated using Sequence Matrix 1.8, and a sequence alignment file for tree building was generated [35]. Modeltest 3.7win, Win-pup4b10 console, and Mrmodeltest2 were used to derive the optimal models for estimating nucleotide substitutions in MrMTgui [36]. The phylogenetic analysis of the *F. fujikuroi* species complex (FFSC) was rooted with *F. sulawesiense*_LC13723; the *F. incarnatum-equiseti* species complex (FIESC) was rooted with *F. concolor*_NRRL13459; the *F. nisikadoi* species complex (FNSC) was rooted with *F. asiaticum*_LC13789; and the *F. sambucinum* species complex (FSAMSC) was rooted with *F. kyushuense*_LC18277; and the phylogenetic tree was constructed using the Bayesian Inference method [37,38].

### 4.11. Morphological Observations

The morphology of the fungal isolates was studied based on the macroscopic and microscopic features [39]. In this study, the isolated strains of *Fusarium* were cultured on SNA and then cultured on PDA and OA to observe the morphological characteristics, growth rates, and pigmentation of pathogenic bacteria [12,39]. One or two strains of each sample of Fusarium were randomly selected to observe the morphological characteristics. The strains were activated from the frozen samples and transferred to a fresh PDA plate for one generation. After 4 days of culture to the second generation, a 5 mm straight cake was prepared by cutting along the colony edge in a concentric circle and transferred to a fresh SNA plate. After 7 days of culture at 25 °C, a 5 mm diameter cake was prepared by cutting along the colony edge in a concentric circle and transferred to a fresh PDA plate and an OA plate. After 3 days of culture at 25 °C, the colony diameter was measured by the crossing method to calculate the growth rate [12]. After 7 days of culture, the colony shape, color, mycelial morphology, and pigmentation were photographed. Five bacterial cakes, each with a diameter of 5 mm, were prepared by cutting along concentric circles of the colony edges and placing them in a 3% mung bean soup medium. After being shaken at 180 rpm at 25 °C for 3 days, the shape of the conidia was recorded, and the sizes of the conidia were measured under a Nikon SMZ25 microscope (Nikon, Shanghai, China). For each species, 50 conidia were randomly measured to calculate the mean value [12].

### 4.12. Assessment of Growth, Spore Production, and Spore Germination on Different Temperatures

The mycelium growth rate method [40] was used to activate the pathogenic bacteria on the PDA medium. The mycelia were placed in a constant temperature incubator at 25 °C. After 4 days of culture, a sterile hole punch (diameter 5 mm) was used to drill holes at the very edges of the growing colony, and a bacteria cake with a diameter of 5 mm was inoculated on new PDA medium. Incubation was carried out at 20, 25, 30, and 35 °C in constant temperature incubators. Each strain had three replicates for each temperature. After 3 days, the colony diameter was measured by the crossing method.

The spore suspension was obtained by static sporulation [41]. The isolates were cultured in constant temperature incubators at 20, 25, 30, and 35 °C for 7 days. Each strain comprised three replicates at each temperature. For each strain, 5 mL of sterile water were added to each petri dish, and the surface of the colony was gently scraped with a slide so that the spores could flow into the sterile water. Three layers of aseptic filter paper were used to filter the liquid and obtain the spore suspension. After shaking, 8 μL of the suspension were absorbed onto a blood cell-counting plate with a pipette, and 5 fields of observation were observed for each strain.

The strains were activated on PDA medium and cultured in a constant temperature incubator at 25 °C for 7 days, and a spore suspension was obtained according to the above method. For spore germination [42], the concentration of spore liquid was adjusted to 1 × 106/mL with sterile water, the clean petri dish was covered with sterilized filter paper, and after wetting the filter paper with sterile water, a slide was placed on the filter paper, and 100 μL water AGAR drops were absorbed on the left and right sides of the slide, and then left for 1 min. The spore suspension (6 μL) was placed onto the water AGAR so that the spore solution was evenly distributed. The strains were cultured in constant temperature incubators at 20, 25, 30, and 35 °C, and 3 replicates were prepared for each temperature. After 3 h, the spore germination value and the total number of spores were observed and recorded. One hundred spores were observed for each strain, and the spore germination rate was calculated as follows: Germination rate (%) = (number of spore germinations/total number of spores observed) × 100 (germination rate estimated to two decimal places). Univariate analysis of variance was performed using the IBM SPSS Statistics 27 software.

### 4.13. Pathogenicity of Rice Spikelet Rot

*F. luffae*, *F. asiaticum*, *F. sulawesiense*, *F. fujikuroi*, and *F. commune* were cultured in the dark for 6 days. Five colony samples with diameters of 5 mm were taken along the edge of the colony and placed into 3% mung bean liquid medium at 25 °C, shaken at 180 rpm for 72 h, and the concentration of spore liquid was adjusted to 1 × 10^6^/mL under the microscope. The spore-liquid spray inoculation method was used to spray fresh suspension onto rice spikelets until water droplets had soaked the surface of the spikelets. The inoculated rice was placed in a temperature-controlled greenhouse under 28 °C, with a light/dark cycle of 12/12 h, and a relative humidity of 80–85%. Furthermore, 5–10 rice plants were inoculated in each treatment, and water was used as a control [15].

## Figures and Tables

**Figure 1 plants-14-01531-f001:**
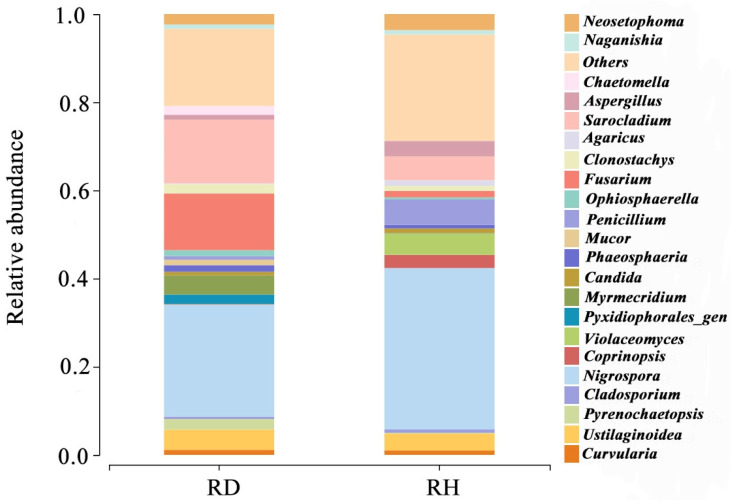
Relative abundances of fungal species in diseased (RD) and relatively healthy (RH) rice spikelets.

**Figure 2 plants-14-01531-f002:**
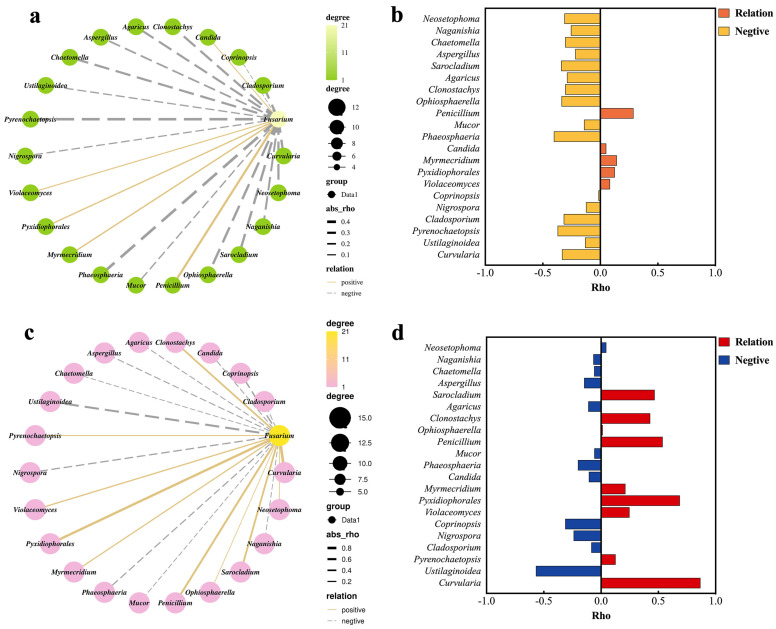
Correlations of *Fusarium* spp. with other fungal groups. (**a**) Network map of the correlations between *Fusarium* and other fungi in relatively healthy rice spikelets (RH); (**c**) network map of correlations between *Fusarium* and other fungi in diseased rice spikelets (RD). The solid lines represent positive correlations, and the dashed lines represent negative correlations. The thickness of the lines indicates the strength of the correlation. *p* value threshold = 1; correlation value thresholds (positive) *p* > 0.5, and (negative) *p* < −0.5. (**b**) Correlations between *Fusarium* spp. and other fungi in relatively healthy rice spikelets (RH); (**d**) correlations between *Fusarium* and other fungi in diseased rice spikelets (RD), |r| > 0.95 is a significant correlation, |r| ≥ 0.8 is highly correlated, 0.5 ≤ |r| ≤ 0.8 is a moderate correlation, and 0.3 ≤ |r| ≤ 0.5 is a low correlation.

**Figure 3 plants-14-01531-f003:**
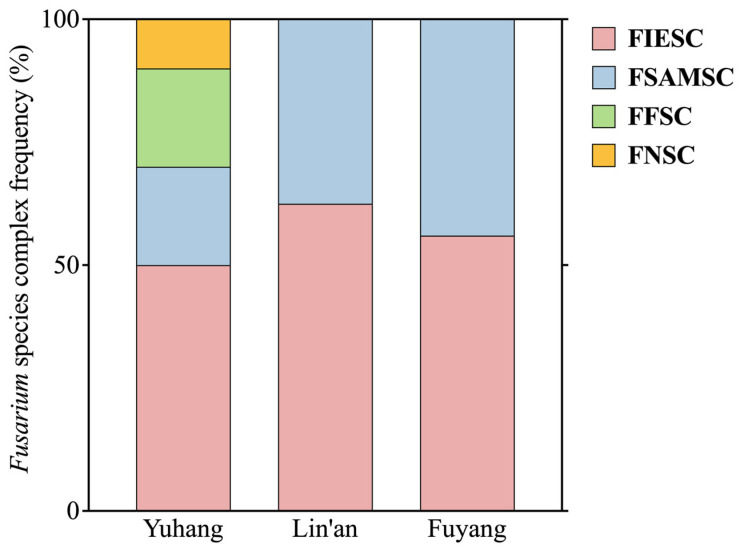
Ratios of *Fusarium* species complex isolates in three regions (Yuhang, Lin’an, and Fuyang). FIESC: *Fusarium incarnatum-equiseti* species complex, FSAMSC: Fusarium sambucinum species complex, FFSC: Fusarium fujikuroi species complex, FNSC: Fusarium nisikadoi species complex.

**Figure 4 plants-14-01531-f004:**
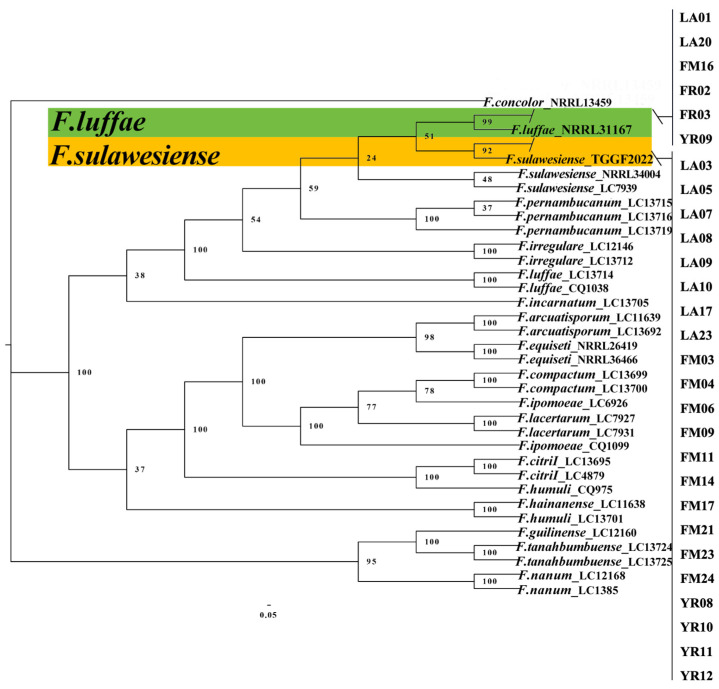
The Bayesian Inference (BI) method was used to construct the phylogeny for *F. sulawesiense* and *F. luffae* based on *tef1*-*rpb1*-*rpb2* multi-sequence concatenation.

**Figure 5 plants-14-01531-f005:**
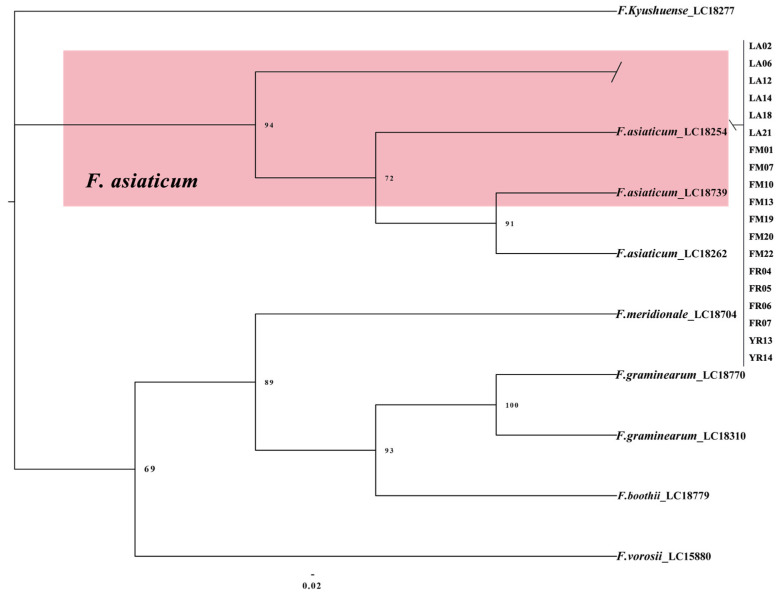
The Bayesian Inference method was used to construct a phylogeny for *F. asiaticum* based on *H3*-*tef1*-*rpb1*-*rpb2* multi-sequence concatenation.

**Figure 6 plants-14-01531-f006:**
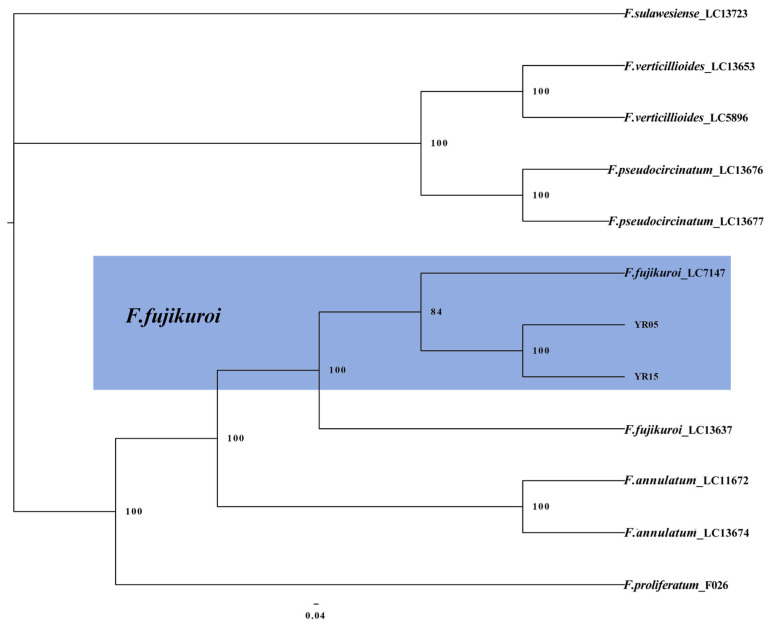
The Bayesian Inference method was used to construct a phylogeny of *F. fujikuroi* based on *CaM*-*tef1*-*rpb1*-*rpb2*-*tub2* multi-sequence concatenation.

**Figure 7 plants-14-01531-f007:**
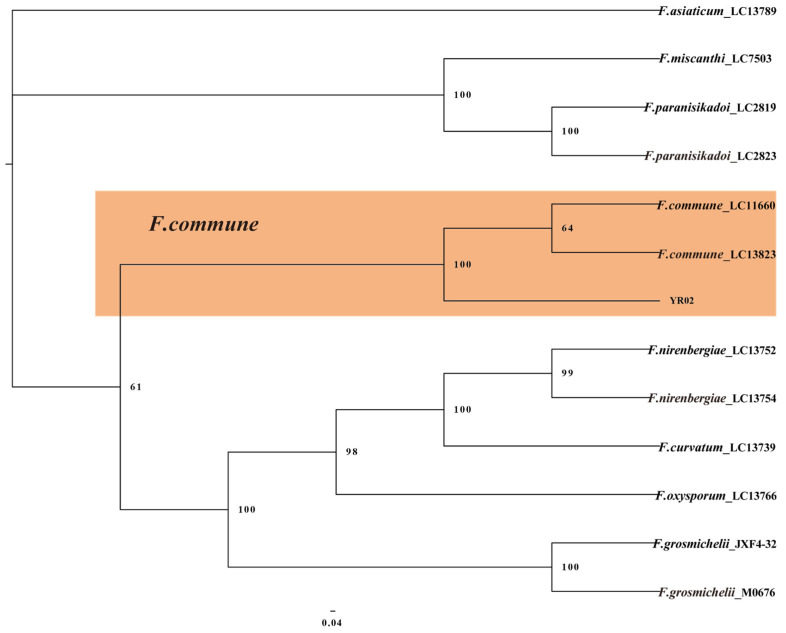
The Bayesian Inference method was used to construct a phylogeny for *F. commune* based on *tef1*-*rpb1*-*rpb2* multi-sequence concatenation.

**Figure 8 plants-14-01531-f008:**
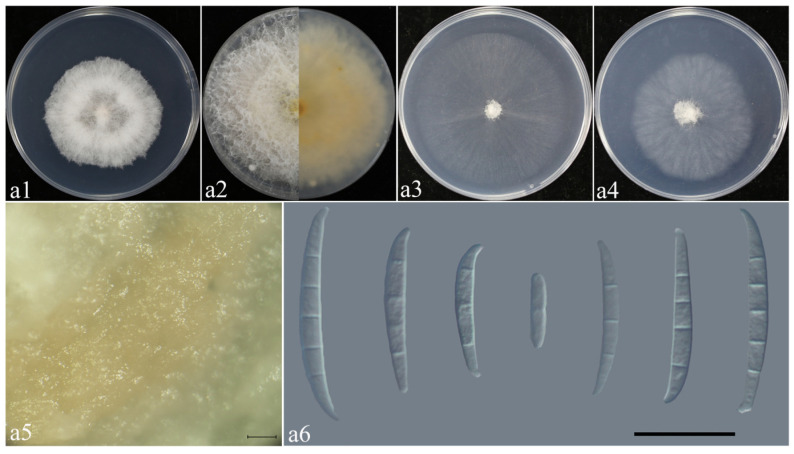
Morphological characteristics of *F. sulawesiense*. (**a1**–**a4**) Colonies on PDA, OA, and SNA; (**a2**) pigmentation; (**a5**) sporodochia. Scale bars: 100 μm. (**a6**) Macroconidia. Scale bars: 20 μm.

**Figure 9 plants-14-01531-f009:**
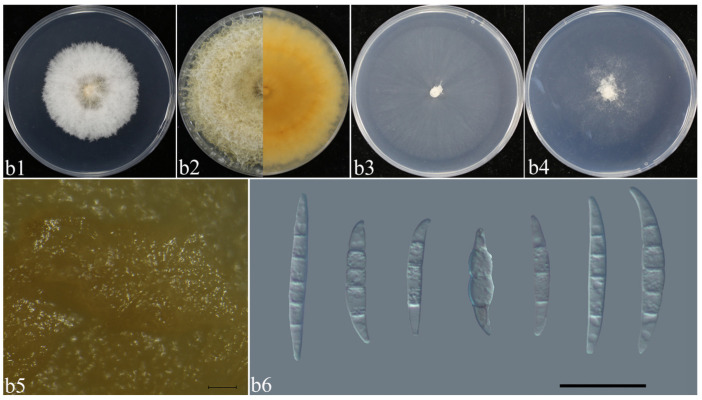
Morphological characteristics of *F. luffae*. (**b1**–**b4**) Colonies on PDA, OA, and SNA; (**b2**) pigmentation; (**b5**) sporodochia. Scale bars: 100 μm. (**b6**) Macroconidia. Scale bars: 20 μm.

**Figure 10 plants-14-01531-f010:**
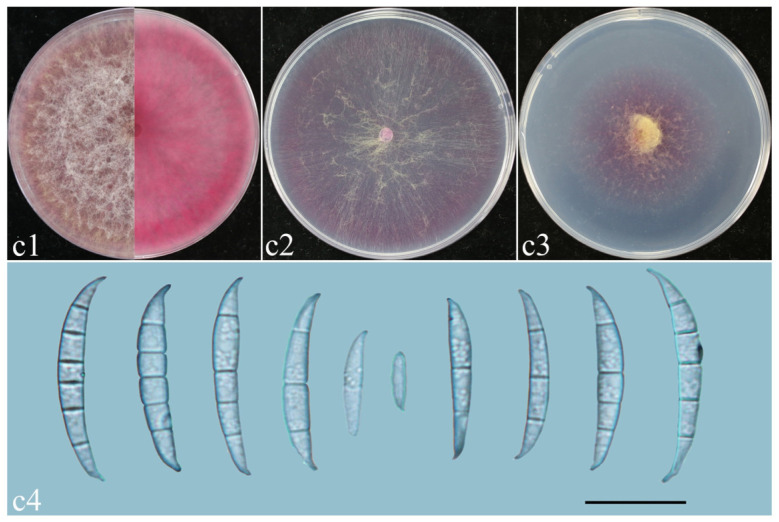
Morphological characteristics of *F. asiaticum*. (**c1**–**c3**) Colonies on PDA, OA, and SNA; (**c4**) macroconidia. Scale bars: 20 μm.

**Figure 11 plants-14-01531-f011:**
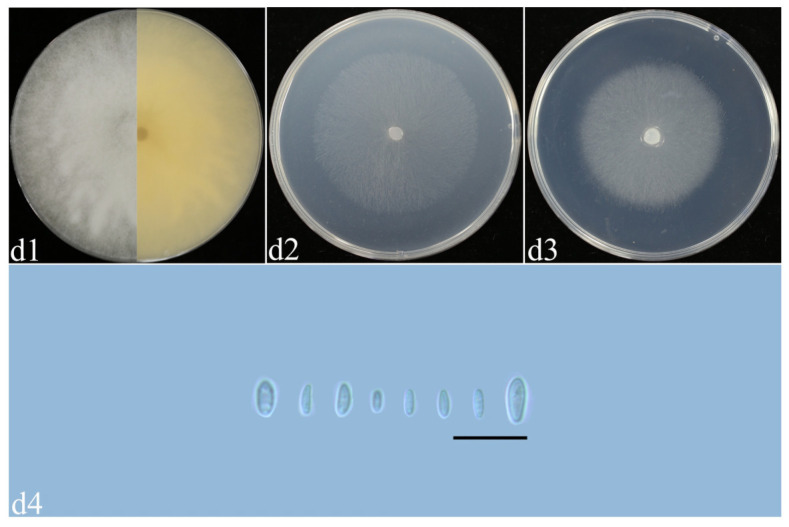
Morphological characteristics of *F. fujikuroi*. (**d1**–**d3**) Colonies on PDA, OA, and SNA; (**d4**) macroconidia. Scale bars: 20 μm.

**Figure 12 plants-14-01531-f012:**
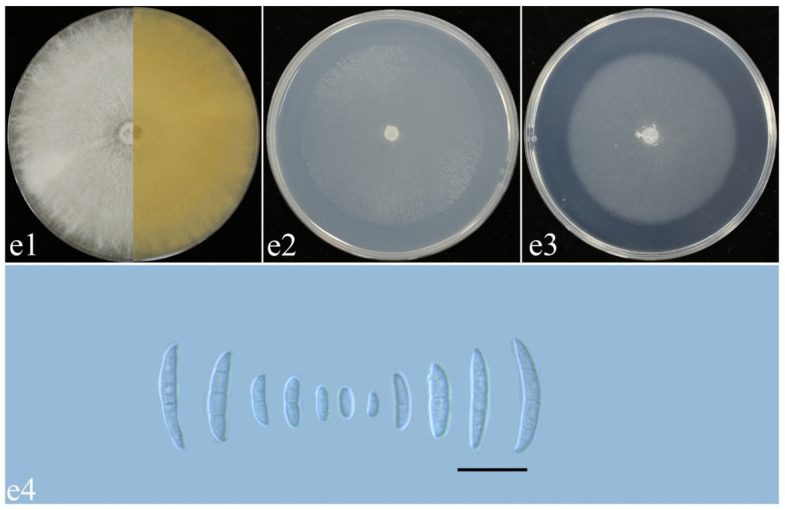
Morphological characteristics of *F. commune*. (**e1**–**e3**) Colonies on PDA, OA, and SNA; (**e4**) macroconidia. Scale bars: 20 μm.

**Figure 13 plants-14-01531-f013:**
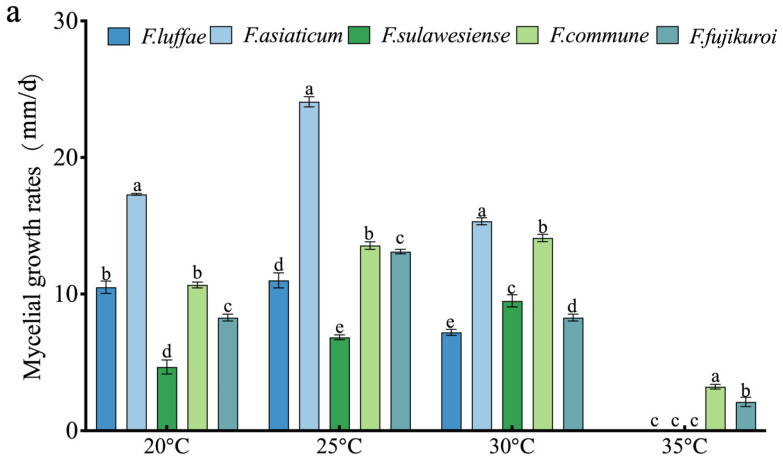
The effects of temperature on the growth, spore production, and spore germination of *Fusarium* of RSRD. (**a**) The growth of *Fusarium* under different temperature conditions. (**b**,**c**) The effects of temperature on the sporulation rate and spore germination rate. Different letters indicate a significant difference (*p* < 0.05), while the same letter indicates no significant difference.

**Figure 14 plants-14-01531-f014:**
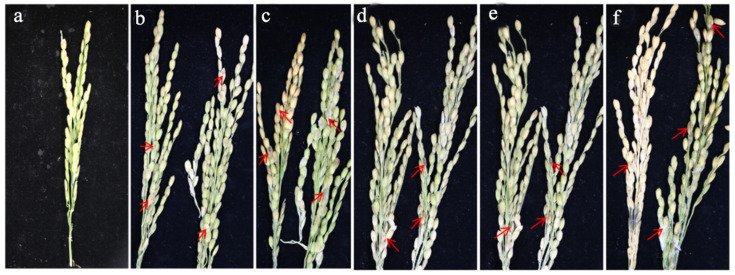
(**a**–**f**) Pathogenicity test of rice spikelet rot. (**a**) CK; (**b**) *F. luffae*; (**c**) *F. asiaticum*; (**d**) *F. sulawesiense*; (**e**) *F. fujikuroi*; (**f**) *F. commune*. The red arrows show the symptoms of RSRD after inoculation with different Fusarium strains.

**Figure 15 plants-14-01531-f015:**
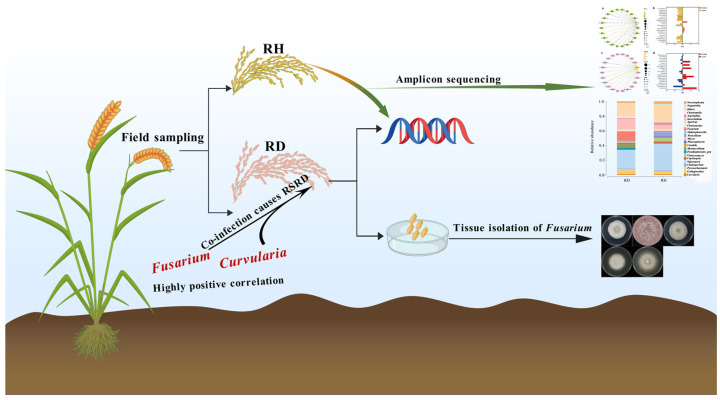
Experimental flowchart of RSRD (from sample collection to data analysis).

**Figure 16 plants-14-01531-f016:**
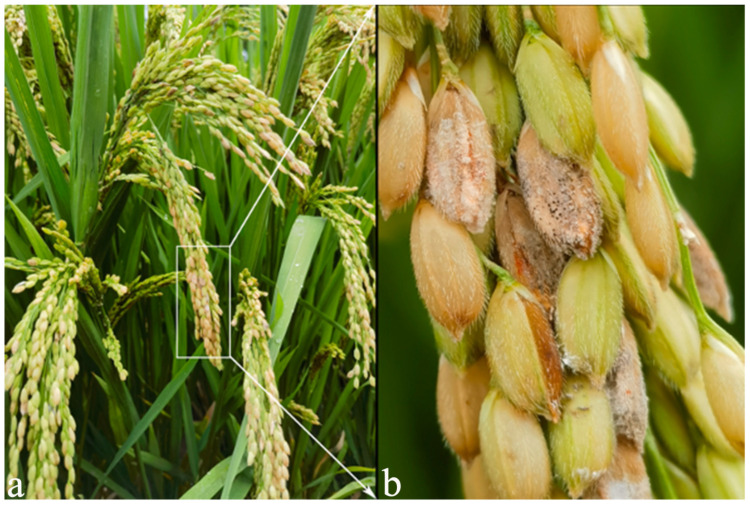
(**a**,**b**) Rice spikelet rot (RSR) caused by multiple species of *Fusarium* showing reddish or discolored glumes, with salmon-colored, white powdery mold, and pink mold layers. Note: the disease name rice spikelet rot (RSR) is adopted in this paper; the disease is also known as Fusarium head blight in the literature [19].

**Table 1 plants-14-01531-t001:** The 51 Fusarium strains isolated from three different areas in Hangzhou that cause rice panicle rot disease obtained in this study.

Species	Strain	Location	GenBank No.					
			*H* _3_	*CAM*	*TEF1*	*RPB1*	*RPB2*	*TUB2*
*F. sulawesiens* (FIESC)	LA03	Lin’an	-	-	PV208419	PV208475	PV208504	-
	LA05	Lin’an	-	-	PV208431	PV208456	PV208485	-
	LA07	Lin’an	-	-	PV208434	PV208450	PV208486	-
	LA08	Lin’an	-	-	PV208420	PV208451	PV208483	-
	LA09	Lin’an	-	-	PV208435	PV208466	PV208487	-
	LA10	Lin’an	-	-	PV208421	PV208465	PV208496	-
	LA17	Lin’an	-	-	PV208422	PV208457	PV208488	-
	LA23	Lin’an	-	-	PV208423	PV208458	PV208498	-
	FM03	Fuyang	-	-	PV208424	PV208459	PV208489	-
	FM04	Fuyang	-	-	PV208436	PV208467	PV208499	-
	FM06	Fuyang	-	-	PV208438	PV208460	PV208484	-
	FM09	Fuyang	-	-	PV208437	PV208472	PV208490	-
	FM11	Fuyang	-	-	PV208430	PV208461	PV208500	-
	FM14	Fuyang	-	-	PV208432	PV208471	PV208491	-
	FM17	Fuyang	-	-	PV208426	PV208452	PV208492	-
	FM21	Fuyang	-	-	PV208433	PV208453	PV208493	-
	FM23	Fuyang	-	-	PV208439	PV208462	PV208494	-
	FM24	Fuyang	-	-	PV208440	PV208468	PV208501	-
	FR01	Fuyang	-	-	PV208441	PV208463	PV208495	-
	YR08	Yuhang	-	-	PV208427	PV208474	PV2084503	-
	YR10	Yuhang	-	-	PV208443	PV208469	PV208497	-
	YR11	Yuhang	-	-	PV208442	PV208455	PV2084502	-
*F. luffae* (FIESC)	LA01	Lin’an	-	-	PV208418	PV208449	PV208476	-
	LA20	Lin’an	-	-	PV208444	PV208473	PV208480	-
	FM16	Fuyang	-	-	PV208445	PV208447	PV208481	-
	FR02	Fuyang	-	-	PV208446	PV208470	PV208482	-
*F. luffae* (FIESC)	FR03	Fuyang	-	-	PV208428	PV208454	PV208479	-
	YR09	Yuhang	-	-	PV208425	PV208448	PV208477	-
	YR12	Yuhang	-	-	PV208429	PV208464	PV208478	-
*F. asiaticum* (FSAMSC)	LA02	Lin’an	PV053218	-	PV053200	PV053237	PV066010	-
	LA06	Lin’an	PV053219	-	PV053201	PV053238	PV066016	-
	LA12	Lin’an	PV053220	-	PV053202	PV053239	PV164744	-
	LA14	Lin’an	PV053221	-	PV164741	PV053240	PV066020	-
	LA18	Lin’an	PV053222	-	PV053203	PV053241	PV066017	-
	LA21	Lin’an	PV053223	-	PV053204	PV066027	PV066011	-
	FM01	Fuyang	PV053224	-	PV053224	PV066028	PV066012	-
	FM07	Fuyang	PV053225	-	PV053207	PV066029	PV066021	-
	FM10	Fuyang	PV053231	-	PV053208	PV164742	PV164743	-
	FM13	Fuyang	PV053232	-	PV053209	PV066030	PV066022	-
	FM19	Fuyang	PV053233	-	PV053210	PV066031	PV066023	-
	FM20	Fuyang	PV053226	-	PV053205	PV066032	PV066024	-
	FM22	Fuyang	PV053234	-	PV053211	PV066033	PV066018	-
	FR04	Fuyang	PV053227	-	PV053212	PV066034	PV066013	-
	FR05	Fuyang	PV053228	-	PV053213	PV066035	PV066014	-
	FR06	Fuyang	PV053229	-	PV053214	PV066036	PV066015	-
	FR07	Fuyang	PV066015	-	PV053215	PV066037	PV066019	-
	YR13	Yuhang	PV053230	-	PV053216	PV066038	PV066025	-
	YR14	Yuhang	PV053236	-	PV053217	PV066039	PV066026	-
*F. fujikuroi* (FFSC)	YR05	Yuhang	-	PQ783814	PQ783818	PQ783816	PQ783815	PQ783820
	YR15	Yuhang	-	PQ783813	PQ783819	PV053199	PQ783817	PQ783821
*F. commune* (FNSC)	YR02	Yuhang	-	-	PQ783810	PQ783811	PQ783812	-

**Table 2 plants-14-01531-t002:** Biological characteristics of *Fusarium* (the mycelial growth rates on PDA, OA, and SNA and the size of microspores).

Species	Growth Rate (mm/d)			Size of Microspore (μm)	
	PDA	OA	SNA	Length	Width
*F. sulawesiense*	6.80 ± 0.81	12.07 ± 0.25	8.33 ± 0.59	29.14 ± 6.96	3.96 ± 0.81
*F. luffae*	11.00 ± 0.55	15.40 ± 0.25	8.87 ± 0.49	28.89 ± 6.29	3.37 ± 0.52
*F. asiaticum*	24.20 ± 0.38	17.13 ± 0.46	10.87 ± 0.46	40.40 ± 21.31	3.43 ± 0.96
*F. fujikuroi*	11.28 ± 0.11	11.56 ± 0.08	11.04 ± 0.10	8.14 ± 1.81	2.89 ± 0.70
*F. commune*	13.76 ± 0.15	11.32 ± 0.10	9.60 ± 0.15	15.15 ± 8.92	3.54 ± 0.99

**Table 3 plants-14-01531-t003:** Composition of the media (Shengong Biology, Shanghai, China).

Medium	Components
PDA 1L	200 g of potato, 20 g of glucose, and 20 g of agar
OA 1L	30 g of oatmeal and 16 g of agar
SNA 1L	1 g of KH_2_PO_4_, 1 g of KNO_3_, 0.5 g of MgSO_4_·7H_2_O, 0.5 g of KCl, 0.2 g of glucose, 0.2 g of sucrose, and 20 g of agar
3% Mung bean soup	30 g of green beans

**Table 4 plants-14-01531-t004:** Amplification procedures and primer information.

Gene	Length	Sequence (5′-3′)	Temperature	References
*tef1*	650 bp	ATGGGTAAGGARGACAAGAC	48 °C	[30]
		GGARGTACCAGTSATCATG		
*CaM*	550 bp	GARTWAAGGAGGCCTTCTC	55 °C	[31]
		TTTTTGCATCATGAGTTGGAC		
*tub2*	680 bp	AACATGCGTGAGATTGTAAGT	55 °C	[19]
		TAGTGACCCTTGGCCCAGTTG		
*rpb1*	1600 bp	CAYAARGARTCYATGATGGGWC	58 °C	[32]
		GTCATYTGDGTDGCDGYTCDCC		
*rpb2*	1700 bp	GGGGWGAYCAGAAGAAGGC	57 °C	[33]
		GCRTGGATCTTRTCRTCSACC		
*H3*	500 bp	ACTAAGCAGACCGCCCGCAGG	60 °C	[34]
		GCGGGCGAGCTGGATGTCCTT		

## Data Availability

Microbial data from this research were submitted to NCBI under the BioProject ID (PRJNA1229133). The data can be accessed via the following link: https://submit.ncbi.nlm.nih.gov/subs/ (This Sequence Read Archive (SRA) submission will be released on 1 June 2025 or upon publication).

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
