# Peer review of "Characterization of *Fusarium* Diversity and Head Microbiota Associated with Rice Spikelet Rot Disease"

_plants, 2025, doi:10.3390/plants14101531_

Round 1

Reviewer 1 Report

Comments and Suggestions for Authors

Abstract

The abstract is informative but too detailed and overlaps with the introduction. It should be more concise. Focus on the key findings, the link between Fusarium and Curvularia, and how temperature influences fungal growth. Avoid listing too many species, and skip citations here. Stick to what was found and why it matters.

Introduction

Lines 37–45 repeat disease symptoms and pathogen lists already mentioned in the abstract.

Lines 72–82 introduce the microbiome but don’t clearly connect it to the study’s goal.

Suggestions:

Remove repeated information and restructure the background to improve flow.

Add a short paragraph explaining what is still unknown—such as how other microbes interact with Fusarium-and how this study addresses those questions using sequencing and microbiome analysis.

Results

Lines 96–100: Taxonomic levels (genus/species) vary; make them consistent.

Lines 116–122: Correlation data is listed but lacks interpretation—explain what these relationships might mean biologically.

Lines 138–139: Mention how many isolates belong to each group and refer briefly to Table S1.

Lines 151–164: Phylogenetic analysis should mention how confident the results are (e.g., bootstrap values).

Figures 8–12: Improve consistency in scale bars and figure labels.

Lines 204–212: Clarify how species were visually distinguished.

Lines 264–266: Say which statistical test was used and include p-values.

Figure 13: Explain the results more clearly and link them to real-world field conditions.

Pathogenicity tests: Use a scoring system to better compare how harmful each species is.

Discussion

Lines 298–301: Calling species “newly discovered” needs proof—like genetic data or formal classification.

Lines 314–318: The link between Fusarium and Curvularia is interesting, but only a correlation. Suggest including a model or figure showing how microbes and the environment might interact to affect disease.

Materials and Methods (Lines 322–554)

Lines 326–336: Sampling details are vague. Add information on how many samples were collected and from where.

Lines 348–363: DNA extraction methods are described, but why these kits were chosen and how DNA quality was checked should be mentioned.

Lines 514–543: The temperature test lacks details on controls and replicates. Also, name the statistical tools and tests used (e.g., ANOVA).

A diagram showing the full study process—from sample collection to data analysis—would improve clarity.

Conclusion

The paper is missing a conclusion section, which is essential. It should briefly restate the main findings (e.g., species diversity, temperature effects, Fusarium–Curvularia link), list study limitations (e.g., small sample area, no functional testing), and suggest future research (e.g., testing biocontrol methods, studying how microbes interact in more detail).

Other Observations

Tables S1 and S2 are referenced but not clearly explained in the main text. Add short descriptions.

“Fusarium paniculae” (Line 267) appears to be a translation error. Use “Fusarium from rice panicles.”

Replace “air mycelium” with “aerial mycelium.”

Make sure all Fusarium species names are italicized.

Use millimeters (mm) consistently instead of mixing with centimeters (cm) for measurements.

Check carefully all the figures, texts, legends, and abbreviations; all are well illustrated and clear and easy to read by the readers.

Author Response

Abstract

The abstract is informative but too detailed and overlaps with the introduction. It should be more concise. Focus on the key findings, the link between Fusarium and Curvularia, and how temperature influences fungal growth. Avoid listing too many species, and skip citations here. Stick to what was found and why it matters.

Response: Thank you for your comments, which I have revised

Introduction

Lines 37–45 repeat disease symptoms and pathogen lists already mentioned in the abstract.

Response: Thank you for your comments, I've removed the duplicates

Lines 72–82 introduce the microbiome but don’t clearly connect it to the study’s goal.

Response:Thank you for your comments, which I have revised

Suggestions:

Remove repeated information and restructure the background to improve flow.

Add a short paragraph explaining what is still unknown—such as how other microbes interact with Fusarium-and how this study addresses those questions using sequencing and microbiome analysis.

Response: Thank you for your comments, (125-133) it has been briefly clarified

Results

Lines 96–100: Taxonomic levels (genus/species) vary; make them consistent.

Response:Thank you for your comments, which I have revised

Lines 116–122: Correlation data is listed but lacks interpretation—explain what these relationships might mean biologically.

Response:Thank you for your comments, which I have revised

Lines 138–139: Mention how many isolates belong to each group and refer briefly to Table S1.

Response:Thank you for your comments, which I have revised

Lines 151–164: Phylogenetic analysis should mention how confident the results are (e.g., bootstrap values).

Response: Thank you for your comments, Bootstrap values have been added

Figures 8–12: Improve consistency in scale bars and figure labels.

Response: Thank you for your comments, in morphological observation, it was taken under different magnifications.

Lines 204–212: Clarify how species were visually distinguished.

Response:Thank you for your comments, which I have added

Lines 264–266: Say which statistical test was used and include p-values.

Response: Thank you for your comments, 4.12(678-679)/Figure 13.(P<0.05) it has been briefly clarified

Figure 13: Explain the results more clearly and link them to real-world field conditions.

Response:Thank you for your comments, which I have revised

Pathogenicity tests: Use a scoring system to better compare how harmful each species is.

Response:We do pathogenicity experiments to detect whether the isolated fungi we pathogen of RSRD.

Discussion

Lines 298–301: Calling species “newly discovered” needs proof—like genetic data or formal classification.

Response: This was changed as: In this study, five species of Fusarium including F. sulawesiense, F. asiaticum, F. luffae, F. commune, and F. fujikuroi were isolated and identified, among which F. asiaticum, F. luffae, and F. commune were firstly determined as pathogens of RSRD.

Lines 314–318: The link between Fusarium and Curvularia is interesting, but only a correlation. Suggest including a model or figure showing how microbes and the environment might interact to affect disease.

Response:OK。

Materials and Methods (Lines 322–554)

Lines 326–336: Sampling details are vague. Add information on how many samples were collected and from where.

Response:Thank you for your comments, which I have revised

Lines 348–363: DNA extraction methods are described, but why these kits were chosen and how DNA quality was checked should be mentioned.

Response: Thank you for your comments, “4.3. Extraction of Microbiome DNA”was sent to BGI for sequencing

Lines 514–543: The temperature test lacks details on controls and replicates. Also, name the statistical tools and tests used (e.g., ANOVA).

Response: Thank you for your comments,‘4.12. Effect of Temperature on Growth, Spore Production, and Spore Germination’has been elaborated——Each strain had three replicates for each temperature and Univariate analysis of variance was performed using IBM SPSS Statistics 27 software.

A diagram showing the full study process—from sample collection to data analysis—would improve clarity.

Response:ok

Conclusion

The paper is missing a conclusion section, which is essential. It should briefly restate the main findings (e.g., species diversity, temperature effects, Fusarium–Curvularia link), list study limitations (e.g., small sample area, no functional testing), and suggest future research (e.g., testing biocontrol methods, studying how microbes interact in more detail).

Response:Thank you for your comments, which I have revised

Other Observations

Tables S1 and S2 are referenced but not clearly explained in the main text. Add short descriptions.

Response:Thank you for your comments,‘Supplementary Materials’ has been elaborated

“Fusarium paniculae” (Line 267) appears to be a translation error. Use “Fusarium from rice panicles.”

Replace “air mycelium” with “aerial mycelium.”

Make sure all Fusarium species names are italicized.

Response:Thank you for your comments, which I have revised

Use millimeters (mm) consistently instead of mixing with centimeters (cm) for measurements.

Response:Thank you for your comments, which I have revised

Check carefully all the figures, texts, legends, and abbreviations; all are well illustrated and clear and easy to read by the readers.

Response:Thank you for your comments, which I have revised

Reviewer 2 Report

Comments and Suggestions for Authors

Latin names of fungal genera and species should be written in Italics. Check and correct it in the whole manuscript including figure captions.

The species identity of the studies strains is not always consistent between the manuscript and data submitted to GenBank. For example, TEF1 sequences of strains LA01 and LA03 were submitted to GenBank under species name Fusarium incarnatum. Therefore, data submitted to GenBank should be checked and corrected if necessary.

Discussion section is relatively short. Consider elaborating on some points. You can for example emphasize the strength of Fusarium species identification carried out in this study (it was based on sequence information form many informative gene regions, not just ITS). The potential of some fungal genera to act as biological control of Fusarium diseases of rice could be discussed based on literature.

Minor comments:

Lines 13 and 190: after “Fusarium isolated from…” a name of host plant is expected, not immediately the name of geographic region. You can write for example: “Fusarium isolated from rice grown in three regions...”

Line 22: Sentence “The occurrence of RSRD contrasted with the traditional perspective” is unclear (perspective on what?)  This sentence should be deleted or rewritten. Many people will read only the abstract, so this text should be understandable on its own.

Line 42: “fullness” does not sound like a correct name of the symptom of RSRD on rice grain. Replace it with more appropriate word.

Line 83: Replace the ambiguous sentence “At present, the mechanisms of RSRD remains unclear” with a more precise description of the problem/gap in research.

Section 2.2 and caption to Fig.2 should state more precisely what kind of variables were correlated with each other. Technically, different fungi were not correlated with each other, but the measures of their abundance in rice spikelets.

Fig. 2a and 2c: names of genera are difficult to read because they are placed over dark circles. Visibility of these names should be improved.

Line 142 and 144: there are wrong references to figure numbers in parentheses.

Figures 4-7: Font size of the text in the phylogenetic trees should be increased because species names are hard to read. The captions contain list of analysis settings written as if they were equations with equal (=) signs. It is better to write proper sentences providing the most important settings for each tree. Settings common for all analyses may be provided in materials and methods.

Figures 8-12 could probably be combined together in one big figure fitting on one page, but the species name would have to appear next to the images. I leave the decision on whether or not to change this to the Authors.

Table 1 should probably be moved under the paragraph where it is first cited. I would add abbreviation of the species complex next to each species. Caption to Tab.1 does not describe well its content. It should make clear that the table contains 51 isolates and the GenBank accession numbers of sequences obtained in three different regions in China. Correct the spelling mistakes in the caption and the first species name.

Caption to Tab.2 should also be written in more precision because term “biological characteristics” is quite vague here. A table caption should explain the table content without the need to check other parts of the manuscript. Provide information on how many representative isolates per species were investigated what values are presented (averages +/- standard deviation?). There are also some Chinese characters in the last line which should be replaced with English words.

Figure 13a: there is a mistake in unit of variable on Y axsis. It should probably be: mm/d.

Line 334: write the full name of BGI.

Line 369: there are two ITS1 primers provided. The second one should probably be ITS2

Author Response

Latin names of fungal genera and species should be written in Italics. Check and correct it in the whole manuscript including figure captions.

Response:Thank you for your comments, which I have revised

The species identity of the studies strains is not always consistent between the manuscript and data submitted to GenBank. For example, TEF1 sequences of strains LA01 and LA03 were submitted to GenBank under species name Fusarium incarnatum. Therefore, data submitted to GenBank should be checked and corrected if necessary.

Response: OK.

Discussion section is relatively short. Consider elaborating on some points. You can for example emphasize the strength of Fusarium species identification carried out in this study (it was based on sequence information form many informative gene regions, not just ITS). The potential of some fungal genera to act as biological control of Fusarium diseases of rice could be discussed based on literature.

Response: Thank you for your comments, which I have revised

Minor comments:

Lines 13 and 190: after “Fusarium isolated from…” a name of host plant is expected, not immediately the name of geographic region. You can write for example: “Fusarium isolated from rice grown in three regions...”

Response:Thank you for your comments, which I have revised

Line 22: Sentence “The occurrence of RSRD contrasted with the traditional perspective” is unclear (perspective on what?)  This sentence should be deleted or rewritten. Many people will read only the abstract, so this text should be understandable on its own.

Response:Thank you for your comments, which I have revised

Line 42: “fullness” does not sound like a correct name of the symptom of RSRD on rice grain. Replace it with more appropriate word.

Response:Thank you for your comments, which I have revised

Line 83: Replace the ambiguous sentence “At present, the mechanisms of RSRD remains unclear” with a more precise description of the problem/gap in research.

Response:Thank you for your comments, which I have revised

Section 2.2 and caption to Fig.2 should state more precisely what kind of variables were correlated with each other. Technically, different fungi were not correlated with each other, but the measures of their abundance in rice spikelets.

Response:Thank you for your comments, which I have revised

Fig. 2a and 2c: names of genera are difficult to read because they are placed over dark circles. Visibility of these names should be improved.

Response: Thank you for your comments, this image cannot be edited.

Line 142 and 144: there are wrong references to figure numbers in parentheses.

Response:Thank you for your comments, which I have revised

Figures 4-7: Font size of the text in the phylogenetic trees should be increased because species names are hard to read. The captions contain list of analysis settings written as if they were equations with equal (=) signs. It is better to write proper sentences providing the most important settings for each tree. Settings common for all analyses may be provided in materials and methods.

Response:OK.

Figures 8-12 could probably be combined together in one big figure fitting on one page, but the species name would have to appear next to the images. I leave the decision on whether or not to change this to the Authors.

Response: Thank you for your comments

Table 1 should probably be moved under the paragraph where it is first cited. I would add abbreviation of the species complex next to each species. Caption to Tab.1 does not describe well its content. It should make clear that the table contains 51 isolates and the GenBank accession numbers of sequences obtained in three different regions in China. Correct the spelling mistakes in the caption and the first species name.

Response:Thank you for your comments, which I have revised

Caption to Tab.2 should also be written in more precision because term “biological characteristics” is quite vague here. A table caption should explain the table content without the need to check other parts of the manuscript. Provide information on how many representative isolates per species were investigated what values are presented (averages +/- standard deviation?). There are also some Chinese characters in the last line which should be replaced with English words.

Response:Thank you for your comments, which I have revised

Figure 13a: there is a mistake in unit of variable on Y axsis. It should probably be: mm/d.

Response:Thank you for your comments, which I have revised

Line 334: write the full name of BGI.

Response:Thank you for your comments, which I have revised

Line 369: there are two ITS1 primers provided. The second one should probably be ITS2

Response:Thank you for your comments, which I have revised

Round 2

Reviewer 1 Report

Comments and Suggestions for Authors

Dear Authors,

Thank you for your detailed response letter. I appreciate the significant improvements made to the manuscript. However, I have several important suggestions to further strengthen your work:

There is an excessive number of abbreviations used throughout the manuscript. Please ensure that all abbreviations are clearly defined at their first appearance, starting from the abstract. A few are still missing initial definitions-double-check the entire manuscript for consistency. I have concerns about the statistical methods used for analyzing the microbiome data. You mentioned using ANOVA, but since microbiome data usually involve relative abundances, non-parametric tests such as PERMANOVA or MANOVA are generally more appropriate. Please confirm that the correct methods were applied and cite relevant references to support your approach. While I cannot fully evaluate the quality of the English, I noticed many small but distracting mistakes (e.g., incorrect use of punctuation, prepositions, and sentence structure). I strongly recommend having the manuscript reviewed by a professional English editor. Figures such as Fig. 2 and Fig. 13 are not clear and do not meet the visual or formatting standards expected in reputable journals. Please revise them to improve clarity, readability, and presentation. 

It is a major issue that your discussion and summary are combined, and there is no distinct conclusion section. In the introduction, you listed four objectives: 1) Understanding the mechanism of RSRD, 2) Examining the distribution of fungal diversity, 3) Exploring the correlation between Fusarium and other fungi, 4) Assessing the influence of Fusarium diversity and temperature on RSRD. Please address each of these objectives clearly and step-by-step in the discussion. Also, compare your findings with relevant recent literature (you already used; you need more to include), and end with a thoughtful conclusion (not a summary of the results) that includes both implications and directions for future research. Thanks for the response letter.

Author Response

  1. There is an excessive number of abbreviationsused throughout the manuscript. Please ensure that all abbreviations are clearly defined at their first appearance, starting from the abstract. A few are still missing initial definitions-double-check the entire manuscript for consistency.

Response: ok

  1. I have concerns about the statistical methodsused for analyzing the microbiome data. You mentioned using ANOVA, but since microbiome data usually involve relative abundances, non-parametric tests such as PERMANOVA or MANOVA are generally more appropriate. Please confirm that the correct methods were applied and cite relevant references to support your approach.

Response:

  1. While I cannot fully evaluate the quality of the English, I noticed many small but distracting mistakes (e.g., incorrect use of punctuation, prepositions, and sentence structure). I strongly recommend having the manuscript reviewed by a professional English editor. 

Response: ok. A native English speaker helped.

  1. Figures such as Fig. 2 and Fig. 13are not clear and do not meet the visual or formatting standards expected in reputable journals. Please revise them to improve clarity, readability, and presentation. 

Response: We tried to improve Fig. 2 and Fig. 13.

It is a major issue that your discussion and summary are combined, and there is no distinct conclusion section. In the introduction, you listed four objectives: 1) Understanding the mechanism of RSRD, 2) Examining the distribution of fungal diversity, 3) Exploring the correlation between Fusarium and other fungi, 4) Assessing the influence of Fusarium diversity and temperature on RSRD. Please address each of these objectives clearly and step-by-step in the discussion. Also, compare your findings with relevant recent literature (you already used; you need more to include), and end with a thoughtful conclusion (not a summary of the results) that includes both implications and directions for future research.

Response: In this journal,  discussion and summary are combined in most manuscripts. This was changed as: As the main pathogen of RSRD, the diversity of Fusarium in the Hangzhou area is different from that in other regions. We found that high temperatures do not completely inhibit the growth of the pathogen. Therefore, when controlling RSRD, targeted measures should be based on the types and characteristics of local pathogens. According to previous studies, the microorganisms associated with rice spike diseases are very diverse. The microbial diversity in diseased rice spikes (RD) is higher than that in healthy ones. We found that Fusarium spp. and Curvularia spp. are positively correlated in RD; however, it is unclear how Fusarium spp.  and Curvularia spp. interact and what role they play in the overall fungal community that causes RSRD. In the future, we can explore the interactions between fungal communities in diseased rice spikes from a microbial perspective.

Round 3

Reviewer 1 Report

Comments and Suggestions for Authors

You said, "In this journal, discussion and summary are combined in most manuscripts. - I do not agree with this response. Editors can handle this issue further.

Response: We tried to improve Fig. 2 and Fig. 13-I dont see any improvement.

Point 2. I have concerns about the statistical methods used for analyzing the microbiome data.
You mentioned using ANOVA ................ you did not respond to this concern.

Author Response

  1. You said, "In this journal, discussion and summary are combined in most manuscripts. - I do not agree with this response. Editors can handle this issue further.

Response: ok.

  1. Response: We tried to improve Fig. 2 and Fig. 13-I dont see any improvement.

Response: We treated Fig. 2 and Fig. 13 in this version.

  1. Point 2. I have concerns about the statistical methods used for analyzing the microbiome data.
    You mentioned using ANOVA ................ you did not respond to this concern.

Response: The following was added in this revised ms.:“The OTU annotation method was used to analyze the species composition and identify the changes and relative abundance of species composition among samples.”

all the revision were tracked in this version.